# Perspectives on Hypoxia Signaling in Tumor Stroma

**DOI:** 10.3390/cancers13123070

**Published:** 2021-06-20

**Authors:** Yuqing Zhang, Morgan Coleman, Rolf A. Brekken

**Affiliations:** 1Hamon Center for Therapeutic Oncology Research, UT Southwestern, Dallas, TX 75390, USA; yuqing.zhang@utsouthwestern.edu (Y.Z.); morgan.coleman@utsouthwestern.edu (M.C.); 2Department of Surgery, UT Southwestern, Dallas, TX 75390, USA; 3Cancer Biology Graduate Program, UT Southwestern, Dallas, TX 75390, USA; 4Division of Pediatric Hematology and Oncology, UT Southwestern, Dallas, TX 75390, USA

**Keywords:** hypoxia, HIF, tumor stroma, immune microenvironment

## Abstract

**Simple Summary:**

Low oxygen tension (hypoxia) caused by high demand of cancer cell proliferation or standard of care therapy is a prevalent feature of solid tumors and is often associated with malignancy. The hypoxia-inducible transcription factor (HIF) family is the critical mediator driving the hypoxia signaling. HIF activity has diverse effects in tumor cells and on tumor stroma, including tumor vasculature, extracellular matrix, fibroblasts and immune cells. In this review, we focus on the effects of HIF in tumor stromal components and discuss essential functions of HIF regulating angiogenesis, collagen deposition and anti-tumor immunity. We also provide a brief overview of the current state of clinical studies targeting tumor hypoxia and provide insights on the limitation of hypoxia-targeted therapies. We believe, with comprehensive knowledge of hypoxia in the tumor microenvironment, challenges of hypoxia-targeted therapies might be better understood and addressed.

**Abstract:**

Hypoxia is a well-known characteristic of solid tumors that contributes to tumor progression and metastasis. Oxygen deprivation due to high demand of proliferating cancer cells and standard of care therapies induce hypoxia. Hypoxia signaling, mainly mediated by the hypoxia-inducible transcription factor (HIF) family, results in tumor cell migration, proliferation, metabolic changes, and resistance to therapy. Additionally, the hypoxic tumor microenvironment impacts multiple cellular and non-cellular compartments in the tumor stroma, including disordered tumor vasculature, homeostasis of ECM. Hypoxia also has a multifaceted and often contradictory influence on immune cell function, which contributes to an immunosuppressive environment. Here, we review the important function of HIF in tumor stromal components and summarize current clinical trials targeting hypoxia. We provide an overview of hypoxia signaling in tumor stroma that might help address some of the challenges associated with hypoxia-targeted therapies.

## 1. Introduction

Low oxygen tension (hypoxia) is prevalent in solid tumors and is associated with disease progression [1]. The growth of cancer cells often exceeds the capacity of the vasculature, and the resulting inadequacy of blood vessel function creates heterogeneous hypoxic areas within the tumor [2]. Hypoxic signaling and the downstream effects are mainly mediated by the hypoxia-inducible transcription factor (HIF) family, which consists of two subunits, HIF-α and HIF-β (also known as the aryl hydrocarbon nuclear translocator, ARNT), of which HIF-β is stably expressed [3]. There are three isoforms of HIF-α—HIF-1α, HIF-2α and HIF-3α, with HIF-1α and HIF-2α having high similarity in structure and being widely studied [4]. HIF-1α is expressed ubiquitously, while HIF-2α and HIF-3α have a more restricted expression pattern [5]. Under normoxic conditions, HIF-α subunits are hydroxylated at proline residues within oxygen-dependent degradation domain (ODD). This prolyl hydroxylation is mediated by prolyl hydroxylase domain-containing enzymes (PHDs) [2,6]. The hydroxylated proline residues (Pro-402/564 in human HIF-α) within ODD domain can be recognized by the Von Hippel–Lindau (VHL) E3 ubiquitin ligase complex, thus mediating HIF-α ubiquitination and proteasomal degradation [6,7]. Under hypoxic conditions, the activity of PHDs is diminished due to lack of oxygen resulting in stabilized HIF-α. Stabilized HIF-α binds to HIF-β to form a heterodimer, which specifically binds hypoxia response elements that drive downstream target gene transcription and facilitate cellular adaptation to hypoxic conditions [8].

In addition to the oxygen deprivation caused by high metabolic demand of cancer cell proliferation, standard of care anticancer therapies can also induce or further exacerbate tumor hypoxia. For example, anti-angiogenic therapy has been reported to increase tumor aggressiveness or lead to therapy resistance in multiple models of cancer [9,10,11]. Antibody-mediated inhibition of vascular endothelial growth factor (VEGF) has been shown to elevate hypoxia in pancreatic tumors and increase collagen deposition, which contributes to tumor aggressiveness [12]. Tyrosine kinase inhibitor BIBF 1120, which targets VEGF receptors, platelet-derived growth factor receptor, and fibroblast growth factor receptor, has also been reported to induce hypoxia in preclinical models of lung and pancreatic cancer [13]. Similar elevations in hypoxia were observed with sorafenib, a standard therapy for hepatocellular carcinoma [14]. In addition, bevacizumab (monoclonal antibody specific for VEGF) in combination with chemotherapy, the current standard of care therapy for metastatic colorectal cancer results in tumor hypoxia, which drives extracellular matrix (ECM) remodeling involved in acquired therapy resistance [15].

Hypoxia in tumors can induce abnormal angiogenesis and desmoplasia, and contribute to the immunosuppressive tumor microenvironment [3]. As a consequence, HIF-1α and HIF-2α expression have been reported to be associated with poor prognosis and metastasis of multiple human cancers [5]. Tumor cells have developed multiple advantages under hypoxic conditions, including increased cell proliferation and migration [16], metabolic changes [17], enhanced cancer cell stemness and resistance to radiotherapies and chemotherapies [18], which have been summarized in multiple reviews [8,19,20]. In addition to cancer cells, hypoxia impacts multiple features of the tumor microenvironment including tumor vasculature, ECM deposition and remodeling and other stromal cells, which may enhance immunosuppression and diminish the efficacy of immunotherapy (Figure 1) [2].

In this review, we will focus on the complex effects of hypoxia and HIF signaling on different cellular and non-cellular compartments in the tumor stroma, including tumor vasculature, ECM, and adaptive and innate immune cells. A current understanding of the hypoxic response of tumor stroma will be provided and current strategies in clinical trials to alleviate tumor hypoxia and the limitation of hypoxia targeted therapy will be discussed.

## 2. Hypoxic Response of Tumor Vasculature

The tumor vasculature is critical for oxygen and nutrient delivery to the cells that make up the tumor microenvironment. However, the tumor vasculature is often dysfunctional, leaky, irregular, and abnormal with aberrant pericytes, which lead to inefficient vessel perfusion [21]. In addition, rapid oxygen consumption in the tumor microenvironment contributes to stabilization of HIF and upregulation of proangiogenic factors such as VEGF secreted from cancer cells and stromal cells, which fuel disorganized new vessel formation [22]. The abnormal tumor vasculature in response to hypoxia can also limit drug delivery and enhance tumor progression and metastasis [22].

Accordingly, multiple studies have been conducted to determine the function of HIF-1α and HIF-2α in endothelial cells. There is a universal switch from HIF-1 to HIF-2 in endothelial cells during the response to hypoxia. HIF-1 activity is initiated in the acute phase of hypoxia while HIF-2 governs the adaptation to prolonged hypoxia [23]. Many factors have been identified that participate in this switch of HIF isoforms, including mRNA stability differences [23,24]. Loss of HIF-1α in Tie2^+^ endothelial cells affects various parameters of endothelial cells, including proliferation, chemotaxis, and wound healing, and causes inhibition of tumor vessel density as well as reduction in tumor growth in a model of lung cancer (Lewis Lung carcinoma (LLC)) [25]. It has been demonstrated that hypoxia induces VEGF, VEGFR1, and VEGFR2 expression on endothelial cells while loss of HIF-1α can block the induction of these genes, thus disrupting the hypoxia-induced VEGF autocrine loop [25]. Consistently, endothelial cell-specific deletion of HIF-1α reduces lung metastasis in a genetic model of breast cancer, MMTV-PyMT, and leads to a reduction of circulating tumor cells in mice bearing LLC tumors [26]. Similarly, HIF-2α deficiency in endothelial cells alters vascular function under physiological conditions and suppresses tumor angiogenesis associated with enhanced hypoxia and increases tumor cell apoptosis in LLC tumors [27]. In addition, in a skin carcinogenesis model, tumors with HIF-2α-deficient endothelial cells exhibit increased numbers of capillaries while reduced numbers of large vessels. However, these small vessels fail to mature into functional blood vessels and lead to perfusion defects, indicating an essential function of HIF-2α in vessel sprouting and remodeling [28]. In summary, these studies indicate compensatory effects of HIF-1α and HIF-2α on tumor angiogenesis, with HIF-1α contributing to vessel growth while HIF-2α responsible for vessel maturation [29].

On the other hand, global heterozygous deficiency of PHD2, the oxygen sensor mediating HIF α subunit hydroxylation and degradation, results in reduced tumor intravasation and metastasis in B16 melanoma, Panc02 pancreatic cancer, and LLC models [30]. These effects were also found in a spontaneous genetic breast cancer model and found to be associated with improved vessel function and maturation [30,31]. These findings indicate the essential functions of HIF signaling in endothelial cells in regulating angiogenesis and highlight some of the potential challenges in targeting HIF activity as an anticancer strategy.

## 3. Hypoxic Response of ECM and CAFs

The extracellular matrix (ECM) is a dynamic collection of noncellular components within tissues including the tumor microenvironment. The ECM consists of collagens, fibronectin, laminin, and other proteoglycans and glycoproteins that contribute to tumor progression [32]. The ECM provides architectural support and tensile strength essential for tissue integrity [33]. In addition, ECM proteins bind to receptors on cancer cells or stromal cells to facilitate cell–ECM adhesion, which can regulate cancer cell proliferation, migration and metastasis [34]. Extensive ECM deposition is generally associated with malignancy of tumor progression as well as impaired drug delivery [35,36]. ECM is also a barrier for T cell infiltration which leads to tumors with a dense ECM typically being T cell-deficient [37]. Collagens, which are mainly produced by fibroblasts, are one of the major components of ECM [34]. Fibroblasts can be activated by cytokines such as TGF-β in the tumor microenvironment and become cancer-associated fibroblasts (CAFs) that gain enhanced capacity to produce and remodel the ECM and are generally considered to promote cancer progression [38,39]. However, a recent study showed that genetic depletion of α-smooth muscle actin (αSMA)-positive CAFs in pancreatic cancer model led to increased immunosuppression and accelerated tumor progression emphasizing the complexity of CAF biology [40,41]. Recently, evidence of CAFs heterogeneity with differential expression of specific markers in distinct subsets of CAFs has emerged, which may help dissect specific functions of CAFs subtypes in tumorigenesis [42,43,44,45].

Hypoxia regulates the homeostasis of ECM through HIF activity as several genes involved in collagen synthesis, modification, and degradation are targets of HIF [46,47]. For example, collagen prolyl 4-hydroxylases (P4Hs), essential enzymes in the synthesis of collagens and collagen lysyl hydroxylases (PLOD2) required for ECM stiffness, are induced in cancer cells by exposure to hypoxia, which is dependent on HIF-1α, but not HIF-2α [48,49]. Elevated expression of P4Hs and PLOD2 correlate with increased cancer cell adhesion to ECM and increased migration [50,51]. Knockdown of PLOD2 in pancreatic stellate cells limits the parallel-patterned fiber architecture formation and suppresses cancer cell directional migration in pancreatic cancer [52]. In non-small cell lung cancer (NSCLC) cell lines, HIF-1α has been shown to form complex with mutant p53 resulting in the specific transcription of ECM genes [53]. Hypoxia has also been shown to elevate the rate of collagen synthesis and deposition in vitro and in vivo [46]. In addition, hypoxia has been shown to induce lysyl oxidase (LOX) and LOX-like proteins (LOXL) secretion by breast cancer cells. LOX and LOXL are amine oxidases that cross-link and stabilize collagen; these enzymes are implicated in remodeling the ECM in metastatic sites to promote metastasis niche formation. LOX/LOXL expression remodels ECM at metastatic sites to facilitate recruitment of bone marrow-derived cells, an effect dependent on HIF-1α and HIF-2α activity in tumor cells [54,55]. As a result, LOX inhibitors are being investigated as potential agents that might overcome chemoresistance and reduce metastasis in triple negative breast cancer [56].

However, the contribution of HIF-1α, and hypoxia by association, in CAFs is less clear. There is evidence that HIF-1α activity in CAFs can function as a tumor promoter. For example, a human fibroblast cell line with active HIF-1α promotes MDA-MB-231 tumor growth when co-injected in vivo with tumor cells. This phenotype is associated with decreased mitochondrial activity and a shift toward aerobic glycolysis [57]. A recent study has confirmed the shift toward lactate and pyruvate production in CAFs isolated from breast cancer patients compared to normal fibroblasts is due to epigenetic reprogramming of HIF-1α and glycolytic enzymes [58]. Chronic hypoxia induces hypomethylation of *HIF1A* promoter as well as promoters of rate-limiting glycolytic genes *PKM* and *LDHA*, which lead to elevated transcript and protein levels and contribute to enhanced glycolytic activity of CAFs derived from breast cancer patients [58]. Furthermore, hypoxic mammary CAFs derived from triple-negative breast cancer patients promote angiogenesis and abnormal vessel formation in a CAF-endothelial cell co-culture system [59]. On the contrary, loss of HIF-1α specifically in FSP1^+^ CAFs was found to accelerate mammary tumor growth and also contribute to decreased tumor vessel density [60]. In addition, global PHD2 haplodeficiency was reported to decrease CAF activation and impair CAF migration and ECM deposition, which reduced metastasis in a spontaneous MMTV-PyMT breast cancer model [31]. Interestingly, the effect on CAF activity relies on PHD2 deletion on tumor cells, but not on CAFs as PHD2 deficiency in platelet-derived growth factor receptor α (PDGFRα)-positive CAFs does not influence metastasis [31]. However, another study provided evidence that depletion of PHD2 in human head and neck CAFs phenocopies the response to hypoxia in a 3D collagen I/Matrigel culture system [61]. Furthermore, a pan-PHD inhibitor (DMOG) reduces tumor stiffness and metastasis in mice bearing 4T1 breast cancer. Interestingly, this efficacy appears to be achieved by targeting PHD2 in CAFs [61]. These contradictory results highlight the complexity and heterogeneity of CAF biology.

In general, the hypoxic tumor microenvironment directly effects collagen deposition and ECM remodeling mainly through HIF activity, which typically enhances tumor progression and metastasis. However, the functions of HIF-1α and HIF signaling in CAFs are more complicated given the heterogeneity of CAF subpopulations, which likely contributes to the seemingly contradictory findings regarding CAF biology and hypoxia.

## 4. The Effect of Hypoxia on T Cells

Hypoxia has direct and complex effects on tumor-infiltrating T cells, including different subtypes of CD4^+^ T helper cells and CD8^+^ effector T cells, potentially resulting in reduced efficacy of immunotherapies.

### 4.1. CD4^+^ T Helper Cells and Regulatory T Cells

There are several subtypes of CD4^+^ T cells, due to divergent differentiation of naïve progenitor cells in response to different cytokine stimuli. The most commonly studied CD4^+^ T cells in the immune response to cancer are T helper (Th)1, Th2, Th17, and regulatory T cell (Treg) [62]. Th1 cells characterized by secreting stimulatory cytokines IFN-γ and TNF-α are considered as proinflammatory and they prime CD8^+^ T cells and are responsible for driving an immune response against tumor cells or infection [63]. While Th2 and Th17 cells may promote tumor growth through expression of immunosuppressive cytokines including IL-4, IL-5, IL-13, and IL-17A, although the contribution of these cells to the tumor immune landscape is not completely clear [62,64,65,66,67,68]. CD4^+^ Tregs are characterized by transcription factor FoxP3 expression and are predominantly immunosuppressive. Tregs maintain peripheral tolerance under normal conditions. The recruitment and expansion of Tregs is enhanced in most tumors and typically impedes antitumor activity of effector cells [69].

Continuous stimulation of the T cell receptor (TCR) under normoxic conditions induces HIF-1α expression through PI3K/mTOR signaling [70,71]. However, hypoxia in tumors can elevate HIF-1α expression in T cells, which contributes to immunosuppression and the differentiation of CD4^+^ T cells. Th1 cells cultured under hypoxia downregulate Th1 responses of IFN-γ production and demonstrate active phosphorylation of STAT3 which leads to further enhanced HIF-1α transcription [72]. In ovarian cancer cells, expression of the chemokine CC-chemokine ligand 28 (CCL28) was reported to be upregulated by hypoxia and correlated with HIF-1α expression. CCL28 promotes tumor growth through the recruitment of CCR10^+^ Tregs [73]. In addition, HIF-1α directly influences the balance between the differentiation of Tregs and Th17 cells by promoting Th17 development while attenuating Treg differentiation through binding FoxP3 resulting in proteasomal degradation [74]. An insertional mutation of FoxP3 that interferes with HIF-1α binding increases FoxP3 stability and Treg differentiation [75]. HIF-1α also contributes to T cell metabolism and thereby promotes lineage choices towards Th17 cells rather than Tregs by favoring glycolytic pathways that are highly active in Th17 cell-inducing conditions [76]. Consistently, another study illustrated the detrimental function of HIF-1α on Treg differentiation by showing that Tregs deficient in VHL, the E3 ubiquitin ligase mediating HIF-1α ubiquitination, lose their suppressive function to control inflammation but produce massive IFN-γ and are converted to Th1-like effector T cells [77]. However, the exact effect of HIF-1α on the differentiation and activity of Tregs remains controversial. Studies have also demonstrated that HIF-1α can promote FoxP3 expression and abundance as well as the suppressive activity of Tregs [78,79,80]. Treg-intrinsic HIF-1α is indispensable for optimal Treg function and Tregs deficient in HIF-1α fail to control inflammation [79]. In addition to studies on HIF-1α, a recent study has demonstrated a critical function of HIF-2α in regulating Treg activity [81]. When HIF-2α was specifically knocked out in Tregs, the cells lost their ability to inhibit effector T cell induced inflammation, which is due in part to upregulated HIF-1α. In tumor settings, HIF-2α-KO Tregs contribute to suppression of MC38 tumor growth and B16F10 melanoma metastasis [81].

Hypoxia is known to stimulate cytokine and chemokine secretion from cancer cells and tumor-infiltrating myeloid cells that recruit Treg infiltration. However, more studies are needed to address the controversial evidence of HIF signaling in determining the differentiation of Tregs and Th17 cells and their subsequent functions.

### 4.2. CD8^+^ Effector T Cells

CD8^+^ effector T cells are the main cytotoxic cell type mediating antitumor immune responses. After antigen-presentation and appropriate co-stimulation from antigen-presenting cells (APCs), CD8^+^ T cells undergo clonal expansion and migration into tumor sites, where they recognize tumor-specific antigens and initiate cytotoxic effects by releasing perforin, granzyme to induce tumor cell apoptosis [82]. The canonical tumor antigen recognition is mainly mediated by αβ CD8^+^ T cells, while there is another distinct population γδ CD8^+^ cells recognizing lipid antigens in an antigen presentation-independent manner [83]. During trafficking into the tumor microenvironment, CD8^+^ T cells encounter a low oxygen tension [84], which can influence their phenotype and activity.

Studies have provided evidence of stimulatory and inhibitory effects of hypoxia on αβ CD8^+^ T cells. Hypoxia can enhance the expression of CD137, a member of the TNF receptor family that functions as costimulatory molecule on activated T cells and is present on tumor-infiltrating lymphocytes (TILs) in multiple implanted and spontaneous tumor models. In HIF-1α-knockout T cells, CD137 expression does not respond to hypoxia and remains negative on those T cells [85]. Furthermore, T lymphocytes that develop under hypoxic conditions are more sustained and lytic with elevated expression of activation markers and enhanced cytokine production [86,87]. Consistently, a later study has demonstrated that enhanced HIF activity by deleting VHL on CD8^+^ T cells increases granzyme B, perforin, and TNF production as well as expression of costimulatory molecules, thus sustaining effector functions that are critical for clearance of viral infection and tumors [84]. Hypoxic cytotoxic T lymphocytes when transferred in vivo were reported to package more granzyme B and be more efficient in controlling tumor growth and improve animal survival in a B16-OVA tumor model [88]. However, loss of HIF-1α but not HIF-2α in CD8^+^ T cells accelerates tumorigenesis [89]. Deletion of HIF-1α or one of it targets, VEGF-A specifically in CD8^+^ T cells limits T cells migration and infiltration into tumor sites in LLC and B16-F10 subcutaneous models [89]. Furthermore, the response of MC38 tumors to anti-PD-1 and anti-CTLA-4 combinational immunotherapy is compromised in mice with HIF-1α loss in CD8^+^ T cells [89]. HIF-1 regulated by mTORC1 is required for glucose metabolism in effector T cells and controls expression of chemokines and adhesion molecules regulating T cell migration [90]. In contrast, hypoxia has also been reported to suppress CD8^+^ T cells. Hypoxia is known to decrease proinflammatory cytokine production from cytotoxic T cells such as IL-2 and IFN-γ and delay T cell development in physiologic oxygen level culture condition [91], which has been shown to be mediated by HIF-1α by specific deletion of HIF-1α in T cells [92]. Hypoxia also increases expression of inhibitory molecules and promotes T cell exhaustion by inducing mitochondrial defects [84,93]. In addition, hypoxia caused by tumor deregulated oxidative metabolism is associated with decreased T cell activity and response to anti-PD-1 therapy [94]. These contradictory results indicate that there might be a balance of HIF-1α activity and the function of cytotoxic T cells in the tumor microenvironment and highlight that these issues are ripe for additional investigation.

## 5. The Effect of Hypoxia on Myeloid Cells

### 5.1. Tumor-Associated Macrophages

As a major component in the tumor microenvironment, tumor-associated macrophages (TAMs) are generally considered as immunosuppressive cells that promote tumor progression, metastasis and angiogenesis [95]. In response to different stimuli in the tumor microenvironment, macrophages are known to exist on a spectrum of phenotypes ranging from immunostimulatory to immunosuppressive. Th1 cell-associated cytokines or LPS stimulation can polarize macrophages into a proinflammatory phenotype (M1-like macrophages), while IL-4 or IL-13 from Th2 cells stimulate an immunosuppressive phenotype (M2-like macrophages) [96,97]. Tumor hypoxia is critical in determining the phenotype of macrophages and hypoxic TAMs have been shown to release factors facilitating tumor growth and immunosuppression [98]. In addition, hypoxia influences the spatial arrangement of macrophages. In more hypoxic regions of a tumor, TAMs tend to be M2-like type while M1-like macrophages are more commonly found in normoxic regions, typically in the periphery of a tumor [98,99]. HIF-1α contributes to the recruitment of bone marrow-derived CD45^+^ myeloid cells including macrophages and further regulates tumor angiogenesis and invasion indirectly [100]. The infiltration of TAMs to hypoxic tumor areas is due to various stimuli including Sema3A/Nrp1 signaling, VEGF, migratory stimulating factors such as colony-stimulating factor 1 (CSF1), CCL2, CCL5 as well as upregulation of TGFβ and M-CSFR [101,102,103]. In addition, a subset of M2-like TAMs (MRC1^+^TIE2^Hi^CXCR4^Hi^) were identified to accumulate around tumor vasculature in breast and lung tumors after chemotherapy, which contributed to tumor relapse after therapy [104].

TAMs in hypoxic regions of tumor tend to be immunosuppressive and have pro-tumor function. Blocking the entry of TAMs into hypoxic regions inhibits their proangiogenic functions and can reverse or reduce the immunosuppressive microenvironment [101]. Specific deletion of HIF-1α in myeloid cells in the MMTV-PyMT model of breast cancer leads to increased tumor cell apoptosis, elevated IFN-γ production by TILs, and delayed tumor progression [105]. In this study, macrophage-mediated T cell suppression under hypoxia was demonstrated to be dependent on HIF-1α and inducible nitric oxide synthetase (iNOS) [105]. In addition, co-culture of macrophages with hypoxic hepatoma cells induces the expression of indoleamine 2,3-dioxygenase (IDO) in macrophages, which also contributes to T cell suppression and Treg expansion [106]. Similarly, loss of HIF-2α in macrophages results in reduced TAM infiltration by reducing the expression of M-CSFR and CXCR4 and improves tumor outcomes in hepatocellular and colon carcinoma models [107]. Besides the effects of hypoxia on TAM infiltration and T cell suppression, hypoxia also induces macrophages to produce increased matrix metalloproteinases (MMPs) including MMP2, 7, and 9, which remodel the ECM and contribute to tumor cell migration and invasion [98,108]. Transcriptome analysis of primary human macrophages cultured under hypoxic conditions identified upregulation of angiogenic factors such as VEGF, cyclooxygenase 2 (COX-2), angiopoietin, and fibroblast growth factor (FGF) [109] Consistently, HIF-1α-deficient macrophages cocultured with tumor spheroids demonstrate enhanced M2-type polarization with attenuated pro-angiogenic properties [110]. Studies have also demonstrated that a small molecule HIF-inhibitor, YC-1 attenuates peritoneal inflammation and proinflammatory M1-type macrophage polarization [111]. However, a recent study has pointed out that bone marrow-derived macrophages from animals with partial loss of the oxygen sensor PHD2 demonstrate more pronounced M2 polarization upon IL-4 stimulation and thus inhibition of PHD2 induces would healing, which indicates the complexity of hypoxia in regulating macrophage polarization [112].

In summary, TAMs tend to infiltrate into hypoxia/necrotic tumor regions, and the hypoxic microenvironment entraps these TAMs and induces an immunosuppressive phenotype, contributing to vessel formation, tumor progression and therapy resistance. Thus, full characterization of TAM plasticity within specific hypoxic and normoxic areas might provide guidance for therapy development and outcome prediction.

### 5.2. Myeloid-Derived Suppressor Cells (MDSCs)

Myeloid-derived suppressor cells (MDSCs) are a group of bone-marrow derived progenitor cells that are considered as precursors of macrophages, dendritic cells, granulocytes, and other mature myeloid cells. MDSCs are generally considered to have immunosuppressive functions in tumors [113]. MDSCs accumulate significantly in tumors and exploit multiple mechanisms to regulate innate and adaptive immune responses [114]. For example, MDSCs promote Treg differentiation and expansion by secreting immunosuppressive cytokines such as IL-10 and TGF-β [114,115]. MDSCs can also produce arginase-1, which depletes arginine, an essential amino acid required for lymphocyte activity. MDSCs also produce reactive oxygen species, which can directly inhibit cytotoxic T cell proliferation and function [113,116].

Hypoxia supports the immunosuppressive function of MDSCs and is a driver of MDSC recruitment [117]. Hypoxia sensing mainly via HIF-1α has been shown to differentiate MDSCs into macrophages and dendritic cells (DCs), thus altering MDSCs function in the tumor site. Similarly, exposure of splenic MDSCs to hypoxia showed similar conversion of MDSCs [118]. In addition, hypoxia culture condition and hypoxic conditioning of liver by cobalt chloride treatment dramatically increase programmed death-ligand 1 (PD-L1) expression on splenic MDSCs isolated from multiple syngeneic tumor models. Mechanistic studies revealed that the upregulation of PD-L1 is regulated by HIF-1α direct binding to the hypoxia response elements (HREs) in the PD-L1 proximal promoter. As a consequence, the suppressive capacity of MDSCs on T cell proliferation is enhanced by hypoxia and blocking the increased PD-L1 expression abrogates MDSC-mediated T cell suppression [119]. HIF-1α can also bind to a HRE in the miR-210 proximal promoter and elevate miR-210 expression, which contributes to enhanced MDSC-mediated T cell suppression [120]. In a recent study in colorectal cancer patients, HIF-1α activity was shown to be associated with V-domain Ig suppressor of T cell activation (VISTA) expression, a negative checkpoint molecule in the B7 family [121]. Subsequent studies in mouse models illustrated that in profound hypoxic regions of tumors, HIF-1α upregulates VISTA expression on tumor-infiltrating MDSCs by direct binding to the promoter, thus promoting the immune suppressive function of MDSCs [121]. Interestingly, hyperoxia (60% oxygen) therapy in the 4T1 triple negative breast cancer model decreases MDSCs expansion and PD-L1 expression in primary and metastatic sites [122]. In general, these studies suggest the enhanced suppressive capacity of MDSCs under hypoxia and demonstrate an additional mechanism by which hypoxia signaling in stromal cells contributes to immunosuppression.

### 5.3. Dendritic Cells

Dendritic cells (DCs) originate from hematopoietic progenitor cells and are a group of professional APCs responsible for T cells priming and initiation of antigen-specific antitumor immune responses [123]. In the absence of environmental stimuli, DCs exist in their immature form with low expression of co-stimulatory molecules and limited capacity for antigen presentation. However, after exposure to bacterial or viral products during infection and proinflammatory cytokines in the tumor microenvironment, DCs undergo maturation and are activated, which allows DCs to present antigens efficiently [123]. Activated DCs express higher levels of major histocompatibility complex (MHC) and costimulatory molecules as well as produce cytokines such as IL-12 and IFNα [124]. However, in the tumor microenvironment, DC maturation and function can be disrupted by production of VEGF, IL-10, IL-6, and decreased co-stimulatory molecule expression [125,126]. In addition, tumor cells are sufficient to convert a subset of immature DCs into TGF-β-secreting regulatory cells that promote the proliferation of Tregs [127].

There are controversial reports regarding the effects of hypoxia on DCs. Hypoxia has been shown to promote the differentiation and migration of DCs through HIF-1α and PI3K/Akt signaling [128,129]. Hypoxia combined with LPS stimulation of DCs leads to upregulated co-stimulatory molecules, enhanced proinflammatory cytokines production and capacity of DCs to stimulate T cells proliferation, which are HIF-1α-dependent [130]. In particular, monocyte-derived DCs generated under hypoxic conditions upregulate triggering receptor expressed on myeloid cells (TREM-1), a hypoxia-induced gene that is responsible for upregulation of costimulatory molecules and secretion of proinflammatory cytokines [131,132,133]. Accordingly, DCs deficient in HIF-1α, when co-cultured with T cells, lead to decreased expression of CD278 and granzyme B in T cells.

In contrast, studies have also demonstrated that in a 3D culture system, hypoxia suppresses maturation of monocyte-derived immature DCs, resulting in decreased motility and phagocytosis [134]. In addition, chronic exposure to hypoxia can induce a cell death program in DCs [135]. Hypoxia can also alter DC phenotype and skew a Th2 polarization of T cells with immunosuppressive cytokine production by upregulating CD44 expression on the surface of DCs [136]. A recent study has indicated that HIF-1α expression in DCs promotes production of immune-inhibitory cytokines and conditional deletion of HIF-1α in DCs enhances their capacity to stimulate T cell response [137]. Furthermore, inhibition of HIF-1α was reported to improve the efficacy of a DC-based vaccine in 4T1 breast cancer model. This was achieved by enhanced cytotoxic T cell proliferation, activity, and IFN-γ production [138]. These contradictory effects of hypoxia on DCs indicate the importance of physiologically appropriate levels of HIF signaling as short-term and prolonged exposure to hypoxia might have different effects on migration and maturation of DCs [128,135].

The effects of HIF signaling in immune cell biology are summarized in Figure 2.

## 6. Targeting Hypoxia and HIFs in Cancer

There are multiple strategies for therapeutically targeting tumor hypoxia including hypoxia activated prodrugs (HAPs), inhibitors of HIF-1, HIF-2, and inhibitors of the associated signaling pathways.

### 6.1. Hypoxia Activated Prodrugs (HAPs)

HAPs are designed to be specifically activated in hypoxic environments and undergo electron reduction to generate an active cytotoxic effector leading to cell death. HAPs are classified by chemical structure and separated into five classes: nitroimidazoles/nitroaromatics, quinones, aromatic n-oxides, aliphatic n-oxides and transition metals [139]. Although there are numerous HAPs that exhibited promising pre-clinical outcomes, the efficacy in clinical trials has been limited, thus far.

Tirapazamine was the first HAP that was developed and tested in clinical trials [140]. It is an aromatic n-oxide compound that is activated to a transient oxidizing radical under hypoxia inducing DNA damage via topoisomerase II [139]. It has been investigated in numerous clinical trials in combination with radiotherapy or cytotoxic chemotherapy for solid tumors including cervical, head and neck, NSCLC, advanced pediatric cancers (NCT00262821, NCT00094081, NCT00006484, NCT00003288) [141]. Overall, the results of phase III studies have been disappointing, and no consistent therapeutic benefit has been identified [142].

TH-302 (evofosfamide), a nitroimidazole, crosslinks DNA in hypoxic tissues, inhibiting cell proliferation and inducing apoptosis [141]. There are extensive preclinical data showing the efficacy of TH-302 as a monotherapy and in combination with standard chemo- or radiotherapy in pancreatic adenocarcinoma, sarcoma, neuroblastoma, and renal cell carcinoma mouse models [143,144,145,146,147,148]. Unfortunately, three phase III clinical trials for pancreatic adenocarcinoma (NCT01746979), soft tissue sarcoma (NCT01440088), and esophageal carcinoma (NCT02598687) were discontinued as no effects were seen [149,150].

There are several additional HAPs (TH-4000, apaziquone, banoxantrone, PR-104) that have been developed and advanced to clinical trials but unfortunately have had similar discouraging results. There are many reviews detailing the preclinical and clinical results of HAPs [141,151].

### 6.2. Inhibitors of HIF

Given that HIF-1 and HIF-2 are highly expressed in a majority of malignancies, inhibitors targeting HIF and HIF signaling are being evaluated widely preclinically and clinically [152].

NLG207 (formerly named CRLX101) is a nanoparticle drug conjugate containing camptothecin, a potent topoisomerase I and HIF-1α inhibitor that accumulates in solid tumors and is slowly released over an extended period of time [153]. NLG207 has demonstrated effective targeting of HIF-1α and angiogenesis in models of breast, prostate cancer, and glioblastoma, as monotherapy or in combination with standard therapies, resulting in inhibition of tumor growth and improved animal survival [154,155,156,157]. In preclinical prostate cancer models, it has recently been demonstrated to improve effects of enzalutamide on previously resistant tumors [158]. It is currently in phase II clinical trials in combination with bevacizumab in ovarian and peritoneal cancer (NCT01652079) [159] and enzalutamide in prostate cancer (NCT03531827) [158].

PT2385 is a direct small molecule inhibitor of HIF-2α. The compound prevents dimerization with HIF-β, inhibiting downstream signaling effects [160]. In preclinical renal cell carcinoma mouse models and patient-derived xenografts, treatment with PT2385 and its analog, PT2399, displays significant suppression of tumor growth, invasion, and angiogenesis [160,161]. It is currently being evaluated in clinical trials for glioblastoma and clear cell renal cell carcinoma (NCT03216499, NCT03108066) [162,163].

Indirect inhibition of HIF can also be achieved through the blockade of associated signaling pathways, such as PI3K/AKT/mTOR and MAPK/ERK pathways. These signaling cascades further activate or enhance HIF-1α synthesis [152]. For example, sirolimus, an mTOR inhibitor has shown encouraging results in preclinical and clinical studies in prostate cancer [164,165]. Furthermore, metformin indirectly inhibits mTORC1 through activation of the AMPK pathway and has also demonstrated promising results [166]. Furthermore, in preclinical models, metformin has been linked to increased T cell activation, working synergistically with checkpoint blockade [167]. It is currently being studied in several ongoing trials for breast, endometrial, colorectal, prostate, and oral cancers (NCT01101438, NCT01697566, NCT02614339, NCT01864096, NCT03685409, NCT02581137) [149]. In addition, Minnelide by targeting p300 and heat shock protein 70 inhibits the transcriptional activity of HIF-1α. Minnelide has shown promising effects in reduction of tumor burden and metastasis in preclinical models and is currently under investigation in a phase II clinical trial for refractory pancreatic cancer (NCT03117920) [168].

### 6.3. Combination with Immune Checkpoint Inhibitors

Although hypoxia targeted therapies have not elicited significant response as a monotherapy and efficacy in combination with standard chemo- and radiotherapy is still under investigation, there are growing evidence to support the combination of hypoxia-targeted strategies with immunotherapy. As previously described, hypoxia induces an immunosuppressive microenvironment by creating dense stroma, inducing barriers to T cell infiltration, increasing MDSCs, and other immunosuppressive myeloid cells [169]. Hypoxic tumors demonstrate few, if any, T cells, and therefore cannot exhibit a robust response to immune checkpoint blockade [170]. Therefore, inhibiting hypoxia might alter the immunosuppressive tumor microenvironment to improve cytotoxic T cell infiltration and function [170]. In transgenic mice bearing prostate adenocarcinoma, combination of TH-302 (evofosfamide) with anti-PD-1/anti-CTLA-4 achieved an 80% cure rate and resulted in an adaptive antitumor response with immune memory [170]. Meanwhile, this combination also increased T cell proliferation, cytotoxic potential and effector cytokine production while mice without evofosfamide were completely resistant to immune checkpoint blockade [170]. Based on these promising results, a phase I trial of TH-302 and ipilimumab (NCT03098160) for advanced solid malignancies is underway [170]. In addition, the effects of hypoxia on the efficacy of immune checkpoint inhibitors, nivolumab and ipilimumab, are being explored in a clinical trial (NCT03003637) [171]. This trial will exploit the use of F-HX4, a hypoxia-specific PET scan radiotracer, to guide biopsies of hypoxic and normoxic tumor tissue. T cell infiltration and effector function pre- and post-immune checkpoint blockade will be compared amongst tumor tissues with varied oxygenation status [171].

### 6.4. Combination with Anti-Angiogenesis Therapy

In addition to combination with immune checkpoint blockade, there is increasing interest in the combination of anti-angiogenic therapies with hypoxia targeted treatments. One mechanism of resistance to anti-angiogenic therapy is through the induction of hypoxia and upregulation of HIF-1α and HIF-2α [172]. Studies have suggested that by adding low dose anti-angiogenic agents, the tumor vasculature could be normalized which should enhance drug delivery [173]. However, it is critical to maintain anti-angiogenic agents at a fine balance as high doses may lead to avascularization of the tumor bed, worsening tumor hypoxia. The combination of TH-302 with anti-angiogenic agents (pazopanib, sunitinib, and DC101) have been evaluated in melanoma, sarcoma, and neuroblastoma mouse models resulting in consistent improved outcomes and decreased tumor growth [174,175,176], leading to the development of phase I/II clinical trials. Furthermore, the combination of TH-302 with bevacizumab is being studied in glioblastoma and high-grade glioma. In a phase II clinical trial, this combination therapy lengthened progression free survival to 4 months in 31% of patients with glioblastoma (NCT02342379) [177]. Another upcoming development is a nanoparticle drug to combine TH-302 (evofosfamide) with an anti-angiogenic/vascular disrupting agent (combretastatin). Combretastatin is bound to the external layer of the nanoparticle and TH-302 is enveloped inside. By releasing combretastatin first from the surface, the tumor vasculature can be normalized, facilitating nanoparticle delivery to the tumor, potentially releasing TH-302 with greater efficacy [178].

Ongoing clinical trials targeting tumor hypoxia have been summarized in Table 1.

## 7. Conclusions

Hypoxia is a common feature of solid tumors and has profound effects on cancer cells and stromal components. Due to tumor heterogeneity, different cell types and tumor components respond differently to hypoxia, highlighting the complexity of this field. Stabilization of HIF under hypoxic conditions upregulates proangiogenic factors and modulates vessel maturation, further contributing to tumor angiogenesis and progression. Hypoxia also promotes ECM remodeling and regulates CAF activation that support tumor growth and metastasis. In addition, hypoxia drives recruitment of immune suppressor cells and modulates phenotype or paracrine factors secretion of immune cells, thereby promoting an immunosuppressive microenvironment which compromises the effects of immunotherapy. However, multiple issues remain to be clarified regarding the specific function of the main sensors of hypoxia. For example, HIF-1α exhibits effects in CAFs, T cells, and DCs in a context-dependent manner, reinforcing the difficulty of targeting hypoxia globally and the importance of developing combination therapies or strategies to overcome the challenges associated with therapeutic targeting of an environmental condition. Combination with immune checkpoint inhibitors and anti-angiogenic therapy is an active investigation and that has potential efficacy in early clinical trials. Moreover, pharmacologic inhibition of PHD enzyme function upregulates HIF signaling and the expression of HIF target genes, which could be applied to treat patients with anemia due to chronic kidney disease and other fibrotic diseases [181]. However, short-term treatment with PHD inhibitors did not show favorable effects on tumor initiation and progression in clinical trials [182]. Interestingly, in a spontaneous breast cancer model, chronic treatment with PHD inhibitors induced erythropoiesis but did not show effects on tumor initiation, progression and metastasis [183]. Further, a pan-PHD inhibitor (DMOG) was reported to limit metastasis in 4T1 breast tumor model [61]. These studies complicated the field of targeting hypoxia. Timing, length of treatment and appropriate disease models are worth considering for optimizing hypoxia-targeted therapies or PHD inhibitors.

Although hypoxia-targeted therapies have achieved efficacy in preclinical models, rare success has been reported in clinical trials. A challenge to this class of targeted therapy is thought to be secondary to narrow therapeutic windows. One strategy being explored to improve treatment response is the development of nanoparticles to allow time-controlled release and drive synergistic effects with other therapies. Another challenge contributing to the failure of prior clinical studies is the fact that the extent of tumor hypoxia has not been evaluated pre or post treatment [171]. Furthermore, previous clinical trials have not stratified patients based on hypoxia status and therefore inadvertently included malignancies that do not exhibit targetable hypoxic conditions. A critical step in improving the efficacy of hypoxia targeted therapies involves optimizing the assessment of hypoxia of individual tumors. There are direct and indirect ways to measure hypoxia (i.e., the placement of electrodes directly in the tumor bed, immunohistochemical staining). However, these modalities have numerous limitations [184]. The most feasible hypoxia assessments are oxygen-enhanced MRI or PET imaging with hypoxia-induced tracers [184]. There are current clinical trials evaluating the sensitivity and specificity of hypoxia targeted imaging (summarized in Table 1). With the challenge of significant intra-tumor heterogeneity, there is a clear unmet need for effective biomarkers to characterize tumor hypoxia. Improved understanding of intratumor heterogeneity of oxygenation and vascularization status would facilitate anti-hypoxia therapy that is tailored based on individual tumor characteristics.

## Figures and Tables

**Figure 1 cancers-13-03070-f001:**
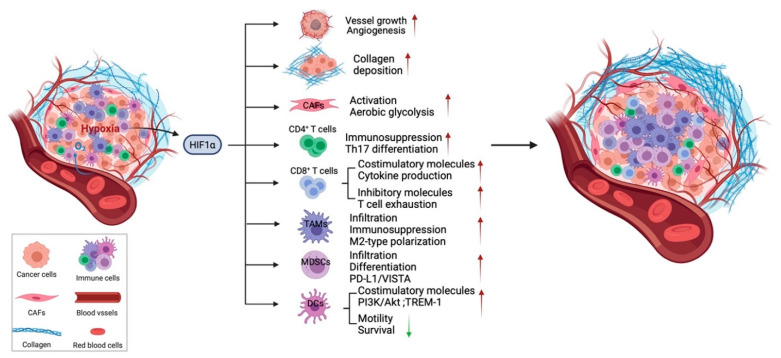
Hypoxia affects the tumor stroma. In addition to multiple effects of hypoxia on cancer cells, hypoxia through HIF-1α signaling regulates the tumor stroma, including tumor vasculature, ECM, CAFs, and immune cells. HIF signaling is essential for vessel growth and maturation. Hypoxia also regulates CAF functions and leads to increased secretion of ECM components. However, the effects of hypoxia on each type of immune cell are complex and, in some cases, controversial. In general, hypoxia results in infiltration of immunosuppressive cells such as TAMs and Tregs and regulates the differentiation, phenotype polarization, as well as the cytotoxic function of immune cells to create an immunosuppressive environment. Hypoxia contributes to tumor progression, metastasis, and compromises the efficacy of standard of care therapy and immunotherapy in numbers of indications. Green arrow pointing down, decreased; red arrow pointing up, increased.

**Figure 2 cancers-13-03070-f002:**
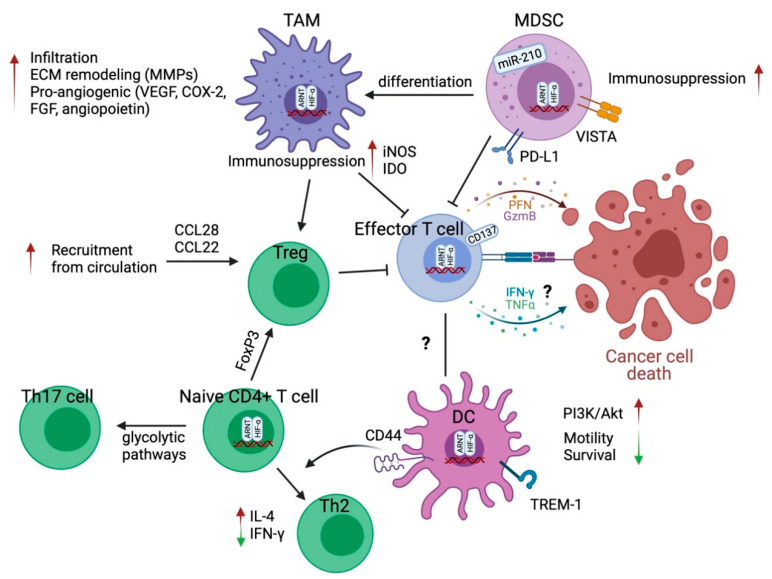
HIF signaling effects on immune cells in the tumor microenvironment. HIF signaling, mainly HIF-1α generally promotes an immunosuppressive microenvironment. In TAMs and MDSCs, HIF-1α contributes to their infiltration and enhanced suppressive capacity on effector T cells. HIF-1α can also stimulate a proangiogenic phenotype of TAMs and TAM-mediated ECM remodeling. The hypoxic microenvironment stimulates the secretion of cytokines and chemokines from tumor cells or TAMs to recruit Tregs, further limiting effector T cell function. There is conflicting evidence of HIF signaling in determining T cell fate. HIF-1α has been demonstrated to induce or inhibit differentiation of naïve CD4^+^ T cells into Th17 T cells or Tregs. HIF-1α signaling also exhibits controversial effects on effector T cell function and the migration and maturation of DCs. The effect of hypoxia on DC and T cell interaction is an area ripe for further investigation. Green arrow pointing down, decreased; red arrow pointing up, increased.

**Table 1 cancers-13-03070-t001:** Ongoing clinical trials targeting and evaluating hypoxia in the tumor microenvironment.

Clinical Trial	Trial Phase	Drug	Mechanism	Disease	References
		**Hypoxia activated prodrug**			
NCT03224182	III	Apaziquone	Indolequinone	Non-muscle invasive bladder cancer	Clinicaltrials.gov
NCT02174549	I/II	Tirapazamine + transarterial embolization	Aromatic n-oxide	Liver cancer	Clinicaltrials.gov
NCT01880359 NCT02661152	III	Nimorazole + chemoradiotherapy	5-nitroimidazoles, radiosensitizer	Locally advanced head and neck squamous cell cancer	Clinicaltrials.gov
		**HIF inhibitors**			
NCT03531827	II	NLG207 + enzalutamide	HIF-1α/topoisomerase I inhibitor + antiandrogen	Metastatic prostate cancer	[158]
NCT02769962 NCT04669002	I/II	EP0057 + olaparib	HIF-1α/topoisomerase I inhibitor + PARP inhibitor	Relapsed/refractory small cell lung cancer Ovarian cancer	Clinicaltrials.gov
NCT03108066 NCT03216499	II	PT2385	HIF-2α inhibitor	Von Hippel-Lindau associated clear cell renal carcinoma Glioblastoma	Clinicaltrials.gov
NCT02974738	I	Belzutifan (PT2977)	HIF-2α inhibitor	Advanced solid tumors	Clinicaltrials.gov
NCT04195750	III	Belzutifan (MK-6482)	HIF-2α inhibitor	Advanced renal cell carcinoma	Clinicaltrials.gov
		**Targeting Associated pathways**			
NCT01101438 NCT01864096 NCT01697566 NCT03685409	III	Metformin	Decreases HIF-1α accumulation	Early-stage breast cancer Low risk prostate cancer Chemoprevention study in endometrial and oral cancer	[149]
NCT02614339	III	Metformin + traditional chemotherapy	Decreases HIF-1α accumulation	Recurrent colorectal cancer	[149]
NCT04275713	II	Metformin + cisplatin	Decreases HIF-1α accumulation	Locally advanced cervical cancer	[179] Clinicaltrials.gov
NCT03117920	II	Minnelide	HSP70, p300 inhibitors	Refractory pancreatic cancer	[149]
NCT03450018	I/II	SLC0111+ gemcitabine	CAIX	Metastatic pancreatic cancer	[180]
NCT04648033 NCT02628080	I	Atovaquone + chemoradiotherapy	Antimalarial drug; Hypoxia modifier via inhibition of mitochondrial complex III	Locally advanced non-small cell lung cancer	[180] Clinicaltrials.gov
		**Combination with immunotherapy or anti-angiogenesis therapy**			
NCT03098160	I	TH-302 + ipilimumab	HAP + anti-CTLA4 Ab	Advanced solid malignancies	[170]
NCT01652079	II	NLG207 + bevacizumab	HIF-1α/topoisomerase I inhibitor + anti-VEGF Ab	Ovarian/peritoneal cancer	[158]
NCT03634540	II	Belzutifan (PT2977) + cabozantinib	HIF-2α + VEGFR2 inhibitors	Advanced clear cell renal carcinoma	Clinicaltrials.gov
NCT04895748	I/Ib	DFF332 + everolimus + spartalizumab	HIF-2α inhibitor + mTOR inhibitor + anti-PD-1 Ab	Relapsed renal cell carcinoma, advanced malignancies with HIF stabilizing mutations	Clinicaltrials.gov
NCT04114136	II	Metformin or rosiglitazone + nivolumab or pembrolizumab	Decreases HIF-1α accumulation	Advanced solid tumor malignancies	Clinicaltrials.gov
		**Assessment of hypoxia**			
NCT03003637	IB/II	18F-FDG PET-CT	Pre and post nivolumab +/- ipilimumab	Advanced/recurrent head and neck carcinoma	[171]
NCT03373994		18F-FDG PET-CT	Evaluate tumor perfusion and hypoxia	Solid tumors	Clinicaltrials.gov
NCT03646747		Oxygen enhanced MRI measurement	Pre and post radiotherapy	Head and neck cancer	Clinicaltrials.gov
NCT04309552		FMISO, FLT PET	Compare FMISO, FLT PET vs. molecular biomarkers of hypoxia and cell proliferation	High grade glioma	Clinicaltrials.gov
NCT02095249		Pimonidazole followed by prostatectomy	Measure tumor hypoxia via immunohistochemical staining	Prostate cancer	Clinicaltrials.gov
NCT04001023		18F-EF5 PET-CT and targeted tumor sampling	Identify molecular differences between hypoxic and non-hypoxic tumors	Advanced ovarian cancer	Clinicaltrials.gov
NCT00568490		Osteopontin, lysyl oxidase, macrophage inhibiting factor and proteomic technology	Identify hypoxic biomarkers in blood and tumors	Head and neck cancer Lung cancer	Clinicaltrials.gov
NCT03054792		18F-FAZA/BOLD PET-MRI	Measure hypoxia between start and completion of treatment	Pediatric sarcomas	Clinicaltrials.gov
		**Hypoxia assessment + radiotherapy**			
NCT04846309	I	FMISO PET + radiation	Hypoxic tumors receive higher dose of radiation	Esophageal cancer	Clinicaltrials.gov
NCT02352792	II	FMISO PET + radiation	Hypoxic tumors receive 10% higher dose of radiation	Head and neck squamous cell carcinoma	Clinicaltrials.gov

HAP: hypoxia activated prodrug; CAIX: carbonic anhydrase IX; FDG: fluorodeoxyglucose; PET: positron emission tomography; FMISO: fluorine-18 fluoromisonidazole; FLT: fluorine-18 fluorothymidine; 18F EF5: fluorine-18 EF5; 18F-FAZA/BOLD: F18-fluoroazomycin arabinoside/blood oxygen level dependent; MRI: magnetic resonance imaging.

## Data Availability

Not applicable.

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
