# Peer review of "Perspectives on Hypoxia Signaling in Tumor Stroma"

_cancers, 2021, doi:10.3390/cancers13123070_

Round 1

Reviewer 1 Report

In this extremely well-written review article Zhang and colleagues give a concise overview on how targeting of the hypoxia-inducible factor (HIF) signaling pathway within the tumor stroma could potentially improve cancer treatment. The authors correctly highlight that the expression of HIF signaling target is increased in several primary tumors. Whether increased HIF activity is required for tumor progression per seor simply a surrogate parameter for malignancy (and growing tumors) is, however, still under debate. So far clinical trials, investigating the role of pharmacologic upregulation of the HIF pathway by so called HIF Prolyl hydroxylase inhibitors to treat, for example, patients with anemia due to chronic kidney disease, have not shown induction of tumor frequencies in patients. Preclinical data likewise show conflicting results regarding the role of HIF during tumor progression (Sangahni and Haase, Adv Chronic Kidney Dis 2019; Strowitzki et al., Pharmacol Res 2019).

Minor concerns:

  1. Whether increased HIF activity is required for tumor progression per seor simply a surrogate parameter for malignancy (and growing tumors) is, however, still unknown and thus topic of constant debate. So far clinical trials, investigating the role of pharmacologic upregulation of the HIF pathway by so called HIF Prolyl hydroxylase inhibitors to treat, for example, patients with anemia due to chronic kidney disease, have not shown induction of tumor frequencies in patients. Preclinical data likewise show conflicting results regarding the role of HIF during tumor progression (Sangahni and Haase, Adv Chronic Kidney Dis 2019; Strowitzki et al., Pharmacol Res 2019). Please discuss the role of HIF and tumor progression in more detail (include, for example, Strowitzki et al., Pharmacol Res 2019).
  2. The authors correctly point out that HIF activity could alter macrophage polarization and that inhibition of HIF-1 or HIF-2 might be useful during cancer therapy altering the macrophage subset. In this context, recent preclinical data suggest that YC-1 in fact specifically inhibits HIF-1 (and not HIF-2) expression during LPS- and hypoxia-inflicted HIF signaling in macrophages and reduces the amount of pro-inflammatory macrophages (Strowitzki et al., Sci Rep 2017, PMID: 29030625). In addition, partial loss of the oxygen sensor PHD2 induced M2 macrophage polarization in vivo and in vitro during intestinal healing which is of particular interest during cancer progression (“never healing wounds”) (PMID: 33784253). Thus, please discuss above-mentioned articles.

Author Response

Response: We appreciate the time and effort that the reviewer has dedicated to providing valuable feedback on our manuscript. We have incorporated changes to reflect most of the suggestions provided by the reviewers. We have highlighted the changes within the manuscript. Please see the attachment. 

Reviewer 2 Report

In general, the article is a reasonable review of hypoxia in tumor and stromal cell biology. Some of the article reads like a list of observations and more effort should be made to provide detail and expand on the concept, rather than just making a brief statement and moving onto the next idea. I have highlighted a few examples below, but a careful re-read by the authors with more emphasis on this suggestion would help improve the article.

The section on immune cell reprogramming by hypoxia and HIFs is well written are reasonably comprehensive. However, the section on targeting hypoxia and HIFs is inadequate. The authors make a reasonable attempt to introduce each targeting strategy but do not provide sufficient examples in each case e.g. HAPs only two are specifically mentioned, HIF-1 inhibitors on NLG207 is mentioned. More examples or a table are needed to make sections 6.1 and 6.2 a more complete review of the field.

Also, I was surprised that no figures were included. The addition of 3-4 figures would significantly improve the review, making it more readable and increasing the chances that the article will be of use to others and this will get cited.

I suggest:

1 figure on HIF and vasculature system changes

1 figure on HIF in CAFs/desmoplasia/metastasis

1 figure of HIF in immune cell biology

1 figure on targeting HIF/hypoxia

Comments and errors

Page 1

Abstract:

Hypoxia “contributes to tumor survival”, this is an unusual statement. While hypoxia acts as a selective pressure for more aggressive cells and promotes progression and metastasis, it seems misleading to state that hypoxia “contributes to tumor survival” when most of the time it would be more likely to contribute to tumor cell death.

“standard-or care therapies” should be standard-of-care therapies

“trials to targeting hypoxia” should be trials targeting hypoxia

Introduction:

Why call hypoxia a hallmark? The term hallmark is associated with certain processes in cancer and should probably be substituted with “feature”.

It is more correct to say “The proline hydroxylated ODD domain can be recognized by the Von Hippel-Lindau…”

“Under hypoxic conditions, the activity of PHDs is diminished due to lack of oxygen resulting in stabilized HIF-α that binds to HIF-β to form a dimer, translocates to the nucleus and specifically binds hypoxia response elements that drive downstream target gene transcription and facilitates cellular adaptation to hypoxic conditions.” This doesn’t read well. Consider breaking into two sentences or fixing the grammar.

“excessive pruning vessels” should read excessive pruning of vessels

Page 2

Define mAb

“contributes” should be contribute

The definition of LLC (Lewis Lung carcinoma) comes after the first use of LLC

Page 3

“On the other hand, global heterozygous deficiency of PHD2, the oxygen sensor mediating HIF α subunit degradation” should state mediating HIF-α subunit hydroxylation and degradation

“challenge and therapeutic potential of targeting HIF activity” should state challenge the therapeutic potential of targeting HIF activity

“upregulated at by HIF-1α” take out at

This paragraph on P4Hs and PLOD2 should also mention the oxygen sensitivity of these ezymes.

“a human fibroblast cell line with active HIF-1α promoted promote MDA” take out promote

Page 4

“A recent study has confirmed the shift toward lactate and pyruvate production in CAFs isolated from breast cancer patients compared to normal fibroblasts due to epigenetic reprogramming of HIF-1α and glycolytic enzymes55.” This statement should include greater detail about the mechanisms implicated, what is the epigenetic programming that is involved?

“In addition, global PHD2 haplodeficiency was reported to decrease CAF activation and impair CAF migration and ECM deposition, which reduced metastasis in a spontaneous MMTV-PyMT breast cancer model.” This sentence needs a reference.

However, another study provided evidence that depletion of PHD2 in human head and neck CAFs phenocopies the response to hypoxia in a 3D collagen I/Matrigel culture system. Furthermore, a pan-PHD inhibitor reduced tumor stiffness and metastasis in mice bearing 4T1 breast cancer. Both of these sentences need references.

“Hypoxia has direct and complex effects on tumor-infiltrating T cells, including different subtypes of CD4+ T helper cells and CD8+ effector T cells, often resulting in reduced efficacy of immunotherapies.” Maybe replace often with potentially

“CD8+ T cells and be responsible for immune response against tumor cells or infection” be should be replaced with are

“The recruitment and expansion of Tregs is enhanced in most tumor and typically impedes antitumor activity of effector cells” should be tumors or tumor types

“insertional mutation of FoxP3 that interfered with HIF-1α binding”

Page 5

“migration into tumor site” site should be sites

“HIF-1α is known to decrease pro-inflammatory cytokines production such as IL-2 and IFN-γ88.” Please elaborate on this comment.

Page 6

“macrophages are known to exit” exit should be exist

“TAMs are tend” take out are

“typically distant from cancer cells94,95.” What does this mean?

“The infiltration of TAMs to hypoxic tumor areas is due to various stimuli including Sema3A/Nrp1 signaling, migratory stimulating factors such as colony-stimulating factor 1 (CSF1), CCL2, CCL5 as well as upregulation of TGFβ and M-CSFR97–99.” What about VEGF?

“hypoxia on TAMs infiltration” TAMS should be TAM

“In summary, TAMs tend to infiltrate into hypoxia/necrotic tumor regions and the hypoxic microenvironment entraps these TAMs and induces an immunosuppressive phenotype, contributing to vessel formation, tumor progression and therapy resistance.” Therapy resistance has not been discussed, please provide more perspective on this idea.

“IL-10, TGF-β107,108” should be IL-10 and TGF-β107,108

Page 7

“macrophages and DCS” what is DCS? If you mean dendritic cells (DCs) please define the abbreviation the first time it is used.

“In a recent study, HIF-1α activity has been shown to be associated with V-domain Ig suppressor of T-cell activation (VISTA) expression, a negative checkpoint molecule in B7 family in colorectal cancer patients”. Needs a reference.

“Antigen-presenting cells (APCs)” has already been defined earlier

“Accordingly, DCs deficient in HIF-1α show decreased expression of CD278 and granzyme B in co-culture with T cells127”. Please confirm, this statement suggests CD278 and granzyme B expression are decreased on the DC cells not the T cells.

Page 8

“one or two electron reduction”. This should simply state one electron reduction for hypoxia. If the authors wish to elaborate on the two electron reduction mechanism they should not it is oxygen independent so does not target hypoxic cells.

“Importantly, when the activated drug diffuses back into normoxic tissues and the cytotoxicity is reversed in the presence of oxygen, limiting toxicity to normal tissues133.” This statement is incorrect the activated cytotoxin can indeed diffuse into normoxic compartments of the tumor but they cannot be oxidized back to the parental HAP once they are fully activated.

“It is currently being investigated in numerous clinical trials in combination with radiotherapy or chemotherapy for solid tumors including cervical, head and neck, NSCLC, advanced pediatric cancers135”. I am not aware of clinical trials testing tirapazamine, these are historical.

“Given that HIF-1 and HIF-2 are expressed highly in a majority of malignancies and the activated downstream PI3K/AKT/mTOR signaling pathway”. This statement suggests that PI3K/AKT/mTOR is downstream of HIF-1 and HIF-2. Please elaborate as to why this is?

Page 10

“Challenges to this class of targeted therapy are thought to be secondary to narrow therapeutic windows in addition to poor tissue penetration.” This statement is overly simplistic, many of the HAPs developed have good tissue penetration properties. As noted by the authors the “failure of prior clinical studies is the fact that the extent of tumor hypoxia has not been evaluated pre or post treatment163.”

Author Response

(The authors gave the same response as above.)

Reviewer 3 Report

In this review article Zhang and colleagues discuss the impact to target hypoxia signaling for cancer therapies in tumor stroma cells and summarize the current ongoing clinical trials in this respect. The review is generally well written and easy to follow. I have some suggestions that need to be taken into account:

1) In the introduction the authors describe the proteasomal degradation of HIF-α subunits. To my knowledge, VHL does not recognize the ODD domain, it recognizes two specific proline residues within the ODD domain of HIF-α subunits. Furthermore, dimerization of HIF-α and HIF-β occurs in the nucleus not in the cytoplasm.

2) Sections 3, 4.1. and 5.3.: The authors should consider too add 2-3 clarifying sentences to the end of each section.

3) On page 4 the authors mentioned a pan-PHD inhibitor. Please specify the inhibitor as there are various PHD inhibitors available that are already in clinical trials or even in clinical use for the treatment of anemia in chronic kidney disease patients. Is anything known about PHD inhibitors in relation to tumor stroma and clinical trials?

4) I would recommend the authors add a tabular overview of ongoing clinical trials to make the review article more visually pleasant to the reader.

Author Response

(The authors gave the same response as above.)

Round 2

Reviewer 2 Report

Would still have appreciated 1 or 2 more figures, but otherwise the authors have address many of the suggestions.